# Predicting Spatial Transcriptomics from Histology Images via Biologically Informed Flow Matching

## Abstract

Spatial transcriptomics (ST) has emerged as a promising technology to bridge the gap between histology imaging and gene expression profiling. However, its application to medical diagnosis is limited due to its low throughput and the need for specialized experimental facilities. To address this issue, we develop STFlow[1], a flow-based generative model to predict spatial transcriptomics from whole-slide histology images. STFlow is trained with a biologically-informed flow matching algorithm that iteratively refines predicted gene expression values, where we choose zero-inflated negative binomial distribution as a prior distribution to incorporate the inductive bias of gene expression data. Compared to previous methods that predict the gene expression of each spot independently, STFlow models the interaction of genes across different spots to account for potential gene regulatory effects. On a recently curated HEST-1k benchmark, we demonstrate STFlow substantially outperforms all baselines including pathology foundation models, with over 18% relative improvement over current state-of-the-art.

## 1 Introduction

Compared to the early days of bulk RNA sequencing, recent advancements in spatial transcriptomics (ST) technology offer a novel approach to molecular profiling within the spatial context of tissues, providing insights into cellular interactions and the microenvironment (Ståhl et al., 2016; Xiao & Yu, 2021). One of the promising clinical applications of ST is the prediction of biomarkers in digital pathology, often visualized in hematoxylin and eosin (H&E)–stained whole-slide images (WSIs), by analyzing the gene expression levels in relation to the tissue morphology (Levy-Jurgenson et al., 2020; Zhang et al., 2022). However, the conventional ST methods (Moffitt et al., 2018; Eng et al., 2019; Ståhl et al., 2016) are low throughput and rely on specialized equipment, limiting their availability compared to standard histology imaging.

To address this, recent works resort to deep learning to predict spatially-resolved gene expression from H&E images. As illustrated in Figure 1(a), a histology image is segmented into small spots, with the objective of predicting the gene expression with the spot image and the coordinate. This line of research has achieved promising results using either an image foundation model to encode local spot-level features (Chen et al., 2024; He et al., 2020; Ciga et al., 2022) or an additional slide-level encoder to incorporate global context (Xu et al., 2024; Chung et al., 2024). However, these methods predict gene expression of each spot independently, thus overlooking the interaction between different genes, i.e. certain genes regulating or influencing the expression of others (Li et al., 2022; Biancalani et al., 2021). To consider such a regulatory effect, we must model the joint distribution over gene expression of all spots in the image, which cannot be solved by single-step regression.

In light of this, we propose STFlow, a flow matching model that casts the original task as a generative modeling problem. As shown in Figure 1(b), the denoiser network of STFlow learns a contextualized representation of each spot that models gene interaction via a novel spatial attention module. Starting from an initial gene expression sampled from the zero-inflated negative binomial (ZINB) distribution (Virshup et al., 2023; Eraslan et al., 2019), STFlow iteratively refines its prediction with a

---

[1]Anonymous codebase: https://anonymous.4open.science/r/Anonymous_STFlow-D420

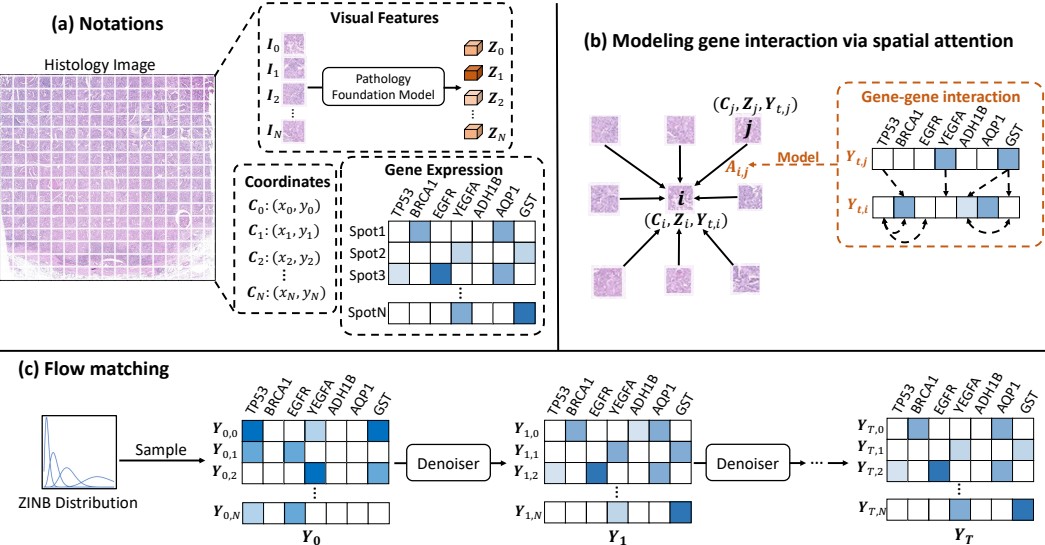

Figure 1: An overview of gene expression prediction from histology image with STFlow. **(a)**: The histology image is segmented into a set of spot images, each associated with a 2D coordinate and gene expression. Each spot image is then encoded using a pathology foundation model. **(b)**: STFlow encodes the slide-level context by aggregating the neighboring spots through spatial attention and the gene-gene interaction is explicitly incorporated within the attention calculation. **(c)**: STFlow iteratively optimizes the gene expression predictions, starting from a sample drawn from the zero-inflated negative binomial (ZINB) distribution.

denoising network (Puny et al., 2021), as illustrated in Figure 1(c). In particular, ZINB distribution as a biologically informed prior allows STFlow to account for the unique nature of gene expression data, offering a more tailored approach than the Gaussian distribution used in standard flow matching.

To validate the effectiveness of STFlow, we evaluate it on the HEST-1k dataset (Jaume et al., 2024), a large-scale collection of ST-WSI pairs comprising 10 benchmarks, and compare its performance against 5 spot-based and 4 slide-based baselines. The experimental results show that STFlow outperforms all baseline approaches and consistently achieves better performance when using visual features extracted by different pathology foundation models, with an average relative improvement of 18%. Additionally, we conduct two case studies on biomarker discovery, where STFlow demonstrates a more significant correlation, highlighting its potential for clinical applications.

## 2 RELATED WORK

**WSI-based spatial gene expression prediction**    Rapid advances in spatial transcriptomics (ST) (Li & Wang, 2021) have enabled the detecting of RNA transcript spatial distribution at sub-cellular resolution. This technology segments hematoxylin and eosin (H&E)–stained whole-slide images (WSIs) into small spots, each providing a corresponding gene expression profile. Conventional ST methods rely on in-situ hybridization techniques(Moffitt et al., 2018; Codeluppi et al., 2018; Eng et al., 2019) or next-generation sequencing approaches (Ståhl et al., 2016; Stickels et al., 2021), which are both costly and time-consuming.

Machine learning-based approaches have recently shown promising results in this domain (Lee et al., 2023). The previous studies fall into two categories: **(1) spot-based approaches** which solely encode the spot and predict the gene expression individually, i.e., modeling $p(\boldsymbol{Y}_i|\boldsymbol{I}_i)^2$ (He et al., 2020; Pang et al., 2021; Chen et al., 2024; Ciga et al., 2022; Xie et al., 2024). Some of these methods leverage foundation models pretrained on large-scale digital pathology datasets, achieving promising results in gene expression prediction (Jaume et al., 2024). **(2) slide-based approaches** which incorporate the slide-level context and predict the gene expression of each spot individually, i.e., modeling

---

[2]We here ignore the time step $t$ and $\boldsymbol{Y}_i$ indicates $i$-th spot's gene expression.

$p(\boldsymbol{Y}_i|\boldsymbol{I}_0, \cdots, \boldsymbol{I}_N)$ (Pang et al., 2021; Zeng et al., 2021; Jia et al., 2024; Xu et al., 2024; Chung et al., 2024). The main idea of these methods is to aggregate the representations of other spots after the image encoders extract each spot's features. The key difference between our proposed STFlow and previous methods is that STFlow explicitly utilizes gene-gene dependency for prediction using a generative model, i.e., modeling joint distribution $p(\boldsymbol{Y}_0, \cdots, \boldsymbol{Y}_N|\boldsymbol{I}_0, \cdots, \boldsymbol{I}_N)$.

**Flow matching** Flow matching is a generative modeling paradigm (Lipman et al., 2022; Albergo & Vanden-Eijnden, 2022; Liu et al., 2022; Jing et al., 2024; Nori & Jin, 2024) that has shown impressive results across various modalities, including images and biomolecules. It defines a sequence of time-dependent probability paths that transform data points from the real distribution to an interpolated sample with a prior distribution. The objective is to approximate the marginal vector field of this path using a neural network. In this work, we repurpose the gene expression regression as a generative task and apply the flow matching since (1) its iterative denoising scheme allows us to incorporate the gene expression within the modeling, and (2) it offers flexibility in selecting a gene expression-specific prior distribution, i.e., zero-inflated negative binomial distribution.

**Geometric deep learning** Geometric deep learning has achieved significant success in chemistry, physics, and biology (Bronstein et al., 2021; Zhang et al., 2023; Liu et al., 2023). The key to this success lies in generating invariant representations for 3D structures, such as molecular conformations, that remain consistent under $E(n)$ transformations, where $n$ represents the dimension of the Euclidean space. $E(n)$ transformations include translations, rotations, and reflections. Previous methods achieve invariance by leveraging invariant features (Satorras et al., 2021; Schütt et al., 2018; Gasteiger et al., 2021) or employing equivariant transformations, such as irreducible representations (Fuchs et al., 2020; Liao & Smidt, 2022; Weiler & Cesa, 2019) and frame averaging (FA) (Puny et al., 2021; Huang et al., 2024). The architecture of the denoiser encodes the spatial context of whole-slide images (WSIs) using an FA-based Transformer architecture, designed to produce invariant representations for each spot, regardless of any $E(2)$ transformations.

## 3 METHOD

In this section, we introduce STFlow, with a biologically informed flow matching denoising framework for leveraging gene interaction and an $E(2)$-invariant denoiser for capturing spatial dependency. We first introduce the necessary background in Section 3.1 and elaborate on the learning framework in Section 3.2. The introduction of architecture is provided in Section 3.3.

### 3.1 PRELIMINARIES

**Problem Formulation** An H&E-stained WSI is segmented into a set of patches, which can be represented as $(\boldsymbol{C}, \boldsymbol{I}, \boldsymbol{Y})$, with coordinates $\boldsymbol{C} \in \mathbb{R}^{N \times 2}$, spot images $\boldsymbol{I} \in \mathbb{R}^{N \times 3 \times H \times W}$, and gene expression levels $\boldsymbol{Y} \in \mathbb{R}^{N \times G}$, where $N$ is the number of spots, $G$ is the number of genes, and $H, W$ indicate the image dimensions. Each element in $\boldsymbol{Y}$ is the count of detected RNA transcripts for a particular gene (starting from 0), representing the gene's expression level. In this study, the goal of STFlow aims to predict the gene expression $\boldsymbol{Y}$ among spots with the input of $(\boldsymbol{C}, \boldsymbol{I})$, which can be formulated as a regression task.

**Pathology Foundation Model** We define $f_{\text{PFM}}(\cdot)$ as a pathology foundation model, which aims to extract general-purpose embeddings for digital pathology after being pretrained on large-scale histology slides, such as Ciga (Ciga et al., 2022), UNI (Chen et al., 2024), and Gigapath (Xu et al., 2024). They receive a patch of the slide as input and produce the embedding for downstream tasks:

$$\{\boldsymbol{Z}_0, \cdots, \boldsymbol{Z}_N\} = f_{\text{PFM}}\left(\{\boldsymbol{I}_0, \cdots, \boldsymbol{I}_N\}\right) \tag{1}$$

where $\boldsymbol{Z}_i, \boldsymbol{I}_i$ represent the $i$-th spot's encoded representation and H&E image. In particular, Gigapath includes a slide encoder that captures the whole-slide context, which we refer to as Gigapath-slide.

In our study, we leverage these foundation models to extract visual features for each spot image instead of training an individual image encoder. The key motivation is that, after being pretrained on large-scale histology slides, these foundation models exhibit strong generalization abilities across different samples and help mitigate batch effects (Jaume et al., 2024).

**Algorithm 1** STFlow: Train

**Require:** Training WSIs $(\boldsymbol{C}, \boldsymbol{I}, \boldsymbol{Y})$
Sample prior $\boldsymbol{Y}_0 \sim \mathcal{Z}(\mu, \phi, \pi)$
Sample timestep $t \sim \text{Uniform}[0, 1]$
Interpolate $\boldsymbol{Y}_t \leftarrow t * \boldsymbol{Y} + (1 - t) * \boldsymbol{Y}_0$
Predict $\hat{\boldsymbol{Y}} \leftarrow f_\theta(\boldsymbol{C}, \boldsymbol{I}, \boldsymbol{Y}_t, t)$
Minimize objective $\text{MSE}(\boldsymbol{Y}, \hat{\boldsymbol{Y}})$

Algorithms 1 and 2 represent the training and inference frameworks of STFlow. The definition of each symbol can be found in the Method section.

**Algorithm 2** STFlow: Inference

**Require:** Testing WSIs $(\boldsymbol{C}, \boldsymbol{I})$
Sample prior $\boldsymbol{Y}_0 \sim \mathcal{Z}(\mu, \phi, \pi)$
**for** $s \leftarrow 0$ **to** $S - 1$
    Let $t_1 \leftarrow s/S$ and $t_2 \leftarrow (s+1)/S$
    Predict $\hat{\boldsymbol{Y}} \leftarrow f_\theta(\boldsymbol{C}, \boldsymbol{I}, \boldsymbol{Y}_{t_1}, t_1)$
    **if** $s = S - 1$ **then**
        **return** $\hat{\boldsymbol{Y}}$
    **end if**
    Interpolate $\boldsymbol{Y}_{t_2} \leftarrow \boldsymbol{Y}_{t_1} + \frac{(\hat{\boldsymbol{Y}} - \boldsymbol{Y}_{t_1})}{(1 - t_1)} * (t_2 - t_1)$
**end for**

## 3.2 LEARNING WITH FLOW MATCHING

Gene interaction is essential for determining the gene expression level. Our key hypothesis is that the expression levels of certain genes in neighboring regions can strongly indicate the target spot's expression (Li et al., 2022; Biancalani et al., 2021; Cordell, 2009). However, this poses a "chicken-and-egg" challenge: the gene-gene dependency we aim to incorporate relies on gene expression as context, which is also what we seek to predict. To address this, we repurpose the gene expression regression model into a generative model, using samples from a prior distribution as input, which is then iteratively optimized instead of performing a one-step prediction.

Specifically, we apply flow matching (Lipman et al., 2022; Albergo & Vanden-Eijnden, 2022) as the optimization framework, which aims to learn a denoised model $f_\theta(\cdot)$:

$$\min_\theta \text{MSE}\left(\boldsymbol{Y}, f_\theta\left(\boldsymbol{Y}_t, \boldsymbol{I}, \boldsymbol{C}, t\right)\right) \tag{2}$$

where $t$ is a time step sampled uniformly from $[0, 1]$, and $\boldsymbol{Y}_t$ is a linear interpolation between $\boldsymbol{Y}$ and a sample $\boldsymbol{Y}_0$ drawn from a prior distribution $p_0(\cdot)$, i.e., $\boldsymbol{Y}_t = t\boldsymbol{Y} + (1 - t)\boldsymbol{Y}_0$. Technically, $f_\theta(\cdot)$ approximates the marginal vector field of the time-dependent conditional probability paths $p_t(\boldsymbol{Y}_t | \boldsymbol{Y})$, allowing it to generate the data $\boldsymbol{Y}$ given the noisy sample from $p_0(\cdot)$.

**Prior Distribution** One of the advantages of flow matching over the diffusion model is its compatibility with different prior distributions. For gene expression data, we apply zero-inflated negative binomial (ZINB) distribution $\mathcal{Z}(\mu, \phi, \pi)$, defined by the following probability mass function:

$$p(y \mid \mu, \phi, \pi) = \begin{cases} \pi + (1 - \pi)\left(\frac{\Gamma(y+\phi)}{\Gamma(\phi)\, y!}\right)\left(\frac{\phi}{\phi+\mu}\right)^\phi \left(\frac{\mu}{\phi+\mu}\right)^y & \text{if } y = 0, \\ (1 - \pi)\left(\frac{\Gamma(y+\phi)}{\Gamma(\phi)\, y!}\right)\left(\frac{\phi}{\phi+\mu}\right)^\phi \left(\frac{\mu}{\phi+\mu}\right)^y & \text{if } y > 0, \end{cases} \tag{3}$$

where $y$ is the count outcome, $\mu$ is the mean of the distribution, $\phi$ denotes the number of failures until stopped, and $\pi$ is the zero-inflation probability. ZINB distribution accounts for the overdispersion and excess zero commonly observed in gene expression data (Virshup et al., 2023; Gayoso et al., 2022; Eraslan et al., 2019).

**Training** As shown in Algo.1, during training, we sample a time step $t$ from the uniform distribution and interpolate the ground-truth gene expression $\boldsymbol{Y}$ with the sampled noise $\boldsymbol{Y}_0$ to obtain noisy sample $\boldsymbol{Y}_t$. The denoiser predicts the denoised gene expression with the inputs of image features, coordinates, noisy samples, and time steps. The model is then optimized by minimizing the difference between the prediction and the ground-truth expression.

**Sampling** As shown in Algo.2, we begin with an initial "expression guess" $\boldsymbol{Y}_0$ sampled from the ZINB distribution and iteratively refine it using the trained denoiser. The model interpolates between the noisy input $\boldsymbol{Y}_t$ and the predicted denoised expression $\hat{\boldsymbol{Y}}$ over multiple steps, with a decay coefficient that gradually increases as the time steps increase. This process ultimately converges to the optimal gene expression in the final step.

### 3.3 DENOISER ARCHITECTURE $f_\theta$

The STFlow's denoiser receives visual features $\boldsymbol{Z}$, coordinates $\boldsymbol{I}$, and gene expression $\boldsymbol{Y}_t$ at time step $t$ as input. The backbone is based on the Transformer architecture (Vaswani, 2017), achieving E(2)-invariance to the coordinates by incorporating frame averaging (FA) within each layer and explicitly encoding spatial dependencies by conducting attention to each spot's local neighbors.

**Local Spatial Context** Cells within the tissues can interact and influence each other's gene expression, thereby forming a spatial context with spot-to-spot dependencies. To efficiently leverage such dependencies, we encode the local spatial context around each spot $i$ and limit the attention to its $k$-nearest neighbors, i.e., $\mathcal{N}(i)$, in the WSI. Long-range context information can be captured through multi-layer attention within the local neighbors of every spot.

**E(2)-Invariant Spatial Attention** We introduce a spatial attention mechanism that generates spot representations invariant to E(2) operations, i.e., rotation, translation, and reflection, of the coordinates. To achieve this, we adapt frame averaging (FA), an E(2)-invariant transformation for point cloud (Puny et al., 2021), to the attention scheme. The flexibility of FA provides a recipe for encoding the coordinates with minimal modification to Transformer. Specifically, for $i$-th spot, we first construct the local context with the direction vectors from it to its neighbors:

$$\mathcal{C}_i = \{\boldsymbol{C}_{i \to j} \mid j \in \mathcal{N}(i)\} \tag{4}$$

where $\boldsymbol{C}_{i \to j} = \boldsymbol{C}_i - \boldsymbol{C}_j$ denotes the direction vector and represents the orientation between spots. Such a geometric context is then projected into multiple frames extracted by PCA:

$$\mathcal{F}(\mathcal{C}_i) := \{(\boldsymbol{U}, \hat{\boldsymbol{c}}) \mid \boldsymbol{U} = [\alpha_1 \boldsymbol{u}_1, \alpha_2 \boldsymbol{u}_2], \alpha_{1,2} \in \{-1, 1\}\}, \tag{5}$$

$$f_{\mathcal{F}}(\mathcal{C}_i) := \{(\boldsymbol{C}_{i \to j} - \hat{\boldsymbol{c}})\boldsymbol{U} \mid (\boldsymbol{U}, \hat{\boldsymbol{c}}) \in \mathcal{F}(\mathcal{C}_i), \boldsymbol{C}_{i \to j} \in \mathcal{C}_i\}$$
$$:= \{\boldsymbol{C}_{i \to j}^{(g)} \mid \boldsymbol{C}_{i \to j} \in \mathcal{C}_i, 1 \le g \le 4\} \tag{6}$$

where $\mathcal{F}(\cdot)$ denotes four extracted frames with the two principal components $(\boldsymbol{u}_1, \boldsymbol{u}_2)$ and centroid $\boldsymbol{c}$, $f_{\mathcal{F}}(\cdot)$ represents the projection of each coordinate using the four extracted frames, and $\boldsymbol{C}_{i \to j}^{(g)}$ denotes the projected direction vector from $i$-th to $j$-th spot using $g$-th frames. Building on top of them, we embed these spatial spot-spot dependencies with linear layers and achieve invariance by averaging the representations in different frames:

$$\boldsymbol{C}'_{i \to j} = \frac{1}{|\mathcal{F}(\mathcal{C}_i)|} \sum_g \mathrm{MLP}(\boldsymbol{C}_{i \to j}^{(g)}) \tag{7}$$

where $\boldsymbol{C}'_{i \to j} \in \mathbb{R}^d$ is the encoded representation of the spatial relationship between $i$-th spot and its neighbor $j$ at $l$-th layer. With such pairwise encoding, the spatial information sent from one source spot depends on the target spot, which is compatible with the attention mechanism. The encoded spatial representation is then incorporated into the attention module.

As shown in Figure 2, the attention module first transforms the image features $\boldsymbol{Z}_i$ into query, key, and value representations:

$$\boldsymbol{Z}_{Q,i} = \boldsymbol{Z}_i \boldsymbol{W}_Q, \ \boldsymbol{Z}_{K,i} = \boldsymbol{Z}_i \boldsymbol{W}_K, \ \boldsymbol{Z}_{V,i} = \boldsymbol{Z}_i \boldsymbol{W}_V \tag{8}$$

where $\boldsymbol{W}_Q, \boldsymbol{W}_K, \boldsymbol{W}_V \in \mathbb{R}^{d \times d}$ are the learnable projections. We adopt MLP attention (Brody et al., 2021) to derive the attention weight between spots, which incorporates the spatial information and the gene expression difference between spots within the calculation:

$$\boldsymbol{A}_{ij} = \mathrm{Softmax}_i \left( \mathrm{MLP} \left( \boldsymbol{Z}_{Q,i} \| \boldsymbol{Z}_{K,j} \| \boldsymbol{C}'_{i \to j} \| (\boldsymbol{Y}_{t,i} - \boldsymbol{Y}_{t,j}) \right) \right) \tag{9}$$

where $\boldsymbol{A}_{ij}$ denotes the attention score between $i$-th and $j$-th spots, and $\mathrm{Softmax}_i(\cdot)$ is the softmax function operated on the attention scores of spot $i$'s neighbors. The spatial representation is then aggregated as the context for updating the spot representation, and the gene

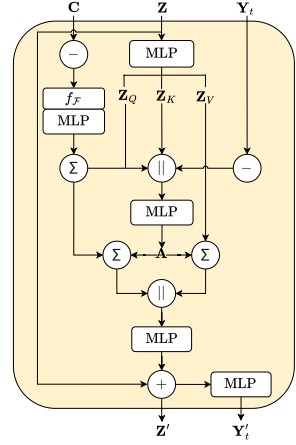

Figure 2: The attention scheme of denoiser.

expression is iteratively updated at each layer, which progressively denoises the gene expression data across different receptive fields:

$$\boldsymbol{Z}'_i = \text{MLP}\left(\sum_{j \in \mathcal{N}(i)} \boldsymbol{A}_{ij}\boldsymbol{Z}_{V,j} \;\|\; \sum_{j \in \mathcal{N}(i)} \boldsymbol{A}_{ij}\boldsymbol{C}_{i \to j}\right) + \boldsymbol{Z}_i \quad \text{and} \quad \boldsymbol{Y}'_{t,i} = \text{MLP}\left(\boldsymbol{Z}'_i\right) \qquad (10)$$

where $\boldsymbol{Z}'_i$ and $\boldsymbol{Y}'_{t,i}$ represent the updated $i$-th spot's representation and gene expression from the spatial attention module. This process is repeated across each layer, with the gene expression updates from each layer averaged to produce the final gene expression prediction.

### 3.4 DISCUSSION

**Notes on invariance** The Equ.7 demonstrates E(2)-invariance to the coordinates as it encodes and averages the coordinates across different frames, which is guaranteed by frame averaging framework. Consequently, the spatial attention mechanism (Equ.9 and Equ.10) that relies on the output of Equ.7 is E(2)-invariant. However, our proposed attention scheme doesn't guarantee the invariance of transformations applied directly to the raw H&E images since we use the embeddings learned by pathology foundation models that are *not* E(2)-invariant.

**Computational Complexity** For spatial attention, FA is efficient due to the low dimensionality of the coordinates (only 2) and the accelerated PCA algorithm, thus we ignore its complexity. The attention calculation involves neighboring spots and linear transformations, resulting in a complexity of $O(Nkd + Nkd^2)$, where $d$ is the embedding size, and is efficient since $k \ll N$. With flow matching, the computation scales linearly with the number of refinement steps $S$. In practice, this remains efficient as flow matching requires relatively few steps, a key advantage over diffusion models. In our experiments, we set $S$ to 5. A wall-clock time comparison can be found in Appendix B.

## 4 EXPERIMENT

In this section, we evaluate our proposed STFlow for gene expression prediction across ten benchmarks and compare its performance against nine baselines. The implementation details can be found in Appendix A, and the dataset statistics can be found in Appendix C.

### 4.1 GENE EXPRESSION PREDICTION

**Datasets** We employ the HEST-1k dataset (Jaume et al., 2024), a large-scale collection comprising spatial transcriptomics data paired with H&E-stained WSIs. Specifically, the dataset includes ten benchmarks[3] covering 48 patients and 74 samples. To prevent data leakage, a patient-stratified split is employed, which results in a $k$-fold cross-validation setup. Following HEST-1k, performance is evaluated using the Pearson correlation between the predicted and measured gene expressions for the top 50 highly variable genes after log1p normalization. We perform cross-validation and report both the mean and standard deviation across the folds.

**Baselines** We compare STFlow with two categories of methods:

- Spot-based approaches, including Ciga (Ciga et al., 2022), UNI (Chen et al., 2024), Gigapath (Xu et al., 2024), STNet (He et al., 2020), and BLEEP (Xie et al., 2024), predict the gene expression solely based on the input spot image. Specifically, BLEEP retrieves the gene expression of spots with similar visual features as prediction. For pathology foundation models, we use a Random Forest model as the regression head, utilizing the visual features extracted by these models, following the setup of HEST-1k.

- Slide-based approaches, including Gigapath-slide, Hist2ST (Zeng et al., 2021), HisToGene (Pang et al., 2021), and TRIPLEX (Chung et al., 2024), incorporate the whole-slide information by

---

[3]Note that the COAD dataset was updated after the paper's release, leading to a significant difference in the performance reported in the HEST-1k manuscript.

aggregating the local or global context around each spot. The coordinates are embedded using a linear layer or a convolution layer, serving as position encoding.

**Results** The comparison results are presented in Table 1, where we also list the image encoder used by each method. It can be observed that, even with a simple linear head, the pathology foundation models demonstrate a significant advantage over most ST-based baselines, which train their image encoders from scratch. However, building on these foundation models, our proposed STFlow can reach better performance and achieve 18% improvement on average, highlighting its compatibility with the pathology foundation models and demonstrating the effectiveness of leveraging spatial context and gene interaction.

Additionally, some ST-based approaches fail to predict significantly correlated gene expression, even with dedicated training on the dataset. We attribute this to the patient-level split, which introduces a more challenging scenario than previous splits, making it difficult for these methods to capture the meaningful semantics of the spot images. This observation is consistent with the findings in Chung et al. (2024). Furthermore, Gigapath-slide, which aggregates whole-slide information, does not outperform Gigapath in these tasks. This may be because the slide encoder's pretrained objective is tailored for slide-level tasks rather than spot-level tasks.

Table 1: Results of gene expression prediction. The image encoder used in each ST-based baseline is listed below each method. The best result is marked in bold, and the best baseline is underlined. OOM indicates an out-of-memory error.

| | Spot-based | | | | | | Slide-based | | | | | |
| | Ciga | UNI | Gigapath | STNet DenseNet121 | BLEEP ResNet50 | Gigapath-slide | Hist2ST ViT | HisToGene ViT | TRIPLEX Ciga | Ciga | STFlow UNI | Gigapath |
|---|---|---|---|---|---|---|---|---|---|---|---|---|
| IDC | $0.423_{.002}$ | $0.502_{.050}$ | $0.514_{.064}$ | $0.380_{.048}$ | $0.346_{.094}$ | OOM | $0.052_{.032}$ | $0.350_{.063}$ | $0.492_{.042}$ | $0.460_{.028}$ | $0.589_{.063}$ | $0.565_{.055}$ |
| PRAD | $0.343_{.001}$ | $0.357_{.000}$ | $0.386_{.008}$ | $0.346_{.006}$ | $0.303_{.004}$ | $0.386_{.006}$ | $0.065_{.038}$ | $0.253_{.005}$ | $0.351_{.023}$ | $0.380_{.001}$ | $0.420_{.005}$ | $0.415_{.013}$ |
| PAAD | $0.406_{.008}$ | $0.424_{.060}$ | $0.436_{.054}$ | $0.370_{.047}$ | $0.347_{.059}$ | $0.394_{.041}$ | $0.111_{.004}$ | $0.303_{.007}$ | $0.429_{.045}$ | $0.440_{.047}$ | $0.506_{.078}$ | $0.513_{.063}$ |
| SKCM | $0.492_{.003}$ | $0.613_{.020}$ | $0.578_{.001}$ | $0.385_{.054}$ | $0.407_{.130}$ | $0.543_{.014}$ | $0.195_{.010}$ | $0.321_{.028}$ | $0.576_{.091}$ | $0.608_{.072}$ | $0.707_{.028}$ | $0.651_{.089}$ |
| COAD | $0.275_{.054}$ | $0.287_{.005}$ | $0.287_{.008}$ | $0.249_{.063}$ | $0.172_{.014}$ | OOM | $0.071_{.006}$ | $0.266_{.015}$ | $0.305_{.004}$ | $0.344_{.023}$ | $0.328_{.013}$ | $0.325_{.023}$ |
| READ | $0.051_{.005}$ | $0.162_{.080}$ | $0.151_{.081}$ | $0.116_{.032}$ | $0.098_{.063}$ | $0.188_{.048}$ | $0.034_{.025}$ | $-0.006_{.013}$ | $0.129_{.062}$ | $0.137_{.075}$ | $0.243_{.002}$ | $0.260_{.023}$ |
| CCRCC | $0.136_{.005}$ | $0.186_{.050}$ | $0.187_{.062}$ | $0.213_{.071}$ | $0.107_{.023}$ | $0.183_{.052}$ | $0.100_{.053}$ | $0.112_{.036}$ | $0.229_{.036}$ | $0.250_{.054}$ | $0.335_{.070}$ | $0.326_{.065}$ |
| HCC | $0.042_{.001}$ | $0.051_{.000}$ | $0.054_{.002}$ | $0.078_{.034}$ | $0.066_{.021}$ | $0.026_{.005}$ | $0.015_{.001}$ | $0.028_{.015}$ | $0.044_{.022}$ | $0.105_{.030}$ | $0.128_{.017}$ | $0.125_{.019}$ |
| LUNG | $0.544_{.001}$ | $0.511_{.030}$ | $0.568_{.038}$ | $0.526_{.025}$ | $0.476_{.021}$ | $0.530_{.025}$ | $0.302_{.063}$ | $0.477_{.057}$ | $0.563_{.036}$ | $0.584_{.027}$ | $0.608_{.021}$ | $0.602_{.013}$ |
| LYMPH | $0.235_{.006}$ | $0.234_{.050}$ | $0.275_{.049}$ | $0.237_{.063}$ | $0.204_{.061}$ | $0.284_{.042}$ | $0.096_{.079}$ | $0.238_{.062}$ | $0.286_{.055}$ | $0.307_{.052}$ | $0.305_{.056}$ | $0.305_{.053}$ |
| Average | 0.305 | 0.347 | 0.344 | 0.290 | 0.252 | / | 0.104 | 0.234 | 0.340 | 0.361 | **0.419** | 0.409 |

## 4.2 FURTHER ANALYSIS ON STFLOW

**Prior distribution comparison** We conduct an experiment to investigate the influence of different prior distributions used in STFlow. Specifically, we replace the ZINB distribution with two alternatives: zero distribution, where all samples are zero, and standard Gaussian distribution.

The results are summarized in Table 2, from which we can observe that the ZINB distribution consistently achieves the best performance across all cases. This demonstrates its effectiveness, as it is better suited to represent gene expression data, which is often sparse and overdispersed. In contrast, the Gaussian distribution fails in certain cases, such as the READ and HCC tasks using Ciga, as it cannot effectively capture the meaningful variation in the non-zero data.

Table 2: Prior distribution comparison on STFlow.

| | | IDC | PRAD | PAAD | SKCM | COAD | READ | CCRCC | HCC | LUNG | LYMPH | Avg. |
|---|---|---|---|---|---|---|---|---|---|---|---|---|
| Ciga | ZINB | $0.460_{.028}$ | $0.380_{.001}$ | $0.440_{.047}$ | $0.608_{.072}$ | $0.344_{.023}$ | $0.137_{.075}$ | $0.250_{.054}$ | $0.105_{.030}$ | $0.584_{.027}$ | $0.307_{.052}$ | 0.361 |
| | Zero | $0.454_{.025}$ | $0.352_{.001}$ | $0.420_{.071}$ | $0.592_{.105}$ | $0.320_{.008}$ | $0.133_{.072}$ | $0.237_{.038}$ | $0.096_{.043}$ | $0.577_{.031}$ | $0.295_{.052}$ | 0.347 |
| | Gaussian | $0.446_{.035}$ | $0.370_{.003}$ | $0.426_{.048}$ | $0.593_{.077}$ | $0.337_{.016}$ | $0.043_{.028}$ | $0.245_{.052}$ | $0.042_{.027}$ | $0.575_{.023}$ | $0.300_{.053}$ | 0.337 |
| Gigapath | ZINB | $0.565_{.055}$ | $0.415_{.013}$ | $0.513_{.063}$ | $0.651_{.089}$ | $0.325_{.023}$ | $0.260_{.023}$ | $0.326_{.065}$ | $0.125_{.019}$ | $0.602_{.013}$ | $0.305_{.053}$ | 0.409 |
| | Zero | $0.564_{.056}$ | $0.411_{.014}$ | $0.506_{.056}$ | $0.651_{.098}$ | $0.323_{.009}$ | $0.261_{.017}$ | $0.328_{.063}$ | $0.115_{.020}$ | $0.593_{.011}$ | $0.301_{.057}$ | 0.405 |
| | Gaussian | $0.559_{.058}$ | $0.403_{.008}$ | $0.507_{.059}$ | $0.643_{.103}$ | $0.320_{.025}$ | $0.252_{.021}$ | $0.320_{.059}$ | $0.115_{.020}$ | $0.594_{.011}$ | $0.297_{.056}$ | 0.401 |
| UNI | ZINB | $0.589_{.063}$ | $0.420_{.005}$ | $0.506_{.078}$ | $0.707_{.028}$ | $0.328_{.013}$ | $0.243_{.002}$ | $0.335_{.070}$ | $0.128_{.017}$ | $0.608_{.021}$ | $0.305_{.056}$ | 0.419 |
| | Zero | $0.585_{.063}$ | $0.397_{.011}$ | $0.494_{.080}$ | $0.686_{.063}$ | $0.321_{.025}$ | $0.234_{.044}$ | $0.324_{.047}$ | $0.103_{.026}$ | $0.608_{.012}$ | $0.291_{.047}$ | 0.404 |
| | Gaussian | $0.580_{.064}$ | $0.409_{.001}$ | $0.498_{.084}$ | $0.677_{.038}$ | $0.309_{.031}$ | $0.213_{.054}$ | $0.316_{.050}$ | $0.116_{.014}$ | $0.600_{.019}$ | $0.288_{.049}$ | 0.400 |

**E(2)-Invariant Architecture Comparison** To demonstrate the effectiveness of our proposed E(2)-invariant denoiser, we implement two representative E(n)-invariant architectures and replace our proposed architecture with them individually:

- EGNN (Satorras et al., 2021) is a representative E($n$) graph neural network that leverages invariant geometric feature distance between coordinates to ensure representation invariance. The model conducts representation aggregation among the $k$-nearest neighbors for each spot. For a fair comparison, EGNN also receives the input of extracted image features as spot features.
- E2CNN (Weiler & Cesa, 2019) is a representative framework for E($n$) convolutional neural networks that utilizes irreducible representations. In our implementation, we use the extracted features as input channels and construct a tensor of neighboring spots centered around the target spots. This tensor is then fed into a ResNet model built using E2CNN.

More details regarding the hyperparameters and implementation can be found in Appendix A. The comparison results are presented in Table 3, where we observe that STFlow's performance decreases to varying degrees when using EGNN or E2CNN as replacements in most cases. We attribute this to the fact that the geometric features used in these models are either simple, as in the case of distances in EGNN, or extracted through constrained functions, such as group steerable kernels in E2CNN. In contrast, FA-based transformation directly leverages the direction vectors, allowing the model to automatically learn relevant geometric features in the latent space.

Table 3: E(2)-invariant architecture comparison.

| | | IDC | PRAD | PAAD | SKCM | COAD | READ | CCRCC | HCC | LUNG | LYMPH | Avg. |
|---|---|---|---|---|---|---|---|---|---|---|---|---|
| Ciga | STFlow | $0.460_{028}$ | $0.380_{001}$ | $0.440_{047}$ | $0.608_{072}$ | $0.344_{023}$ | $0.137_{075}$ | $0.250_{054}$ | $0.105_{030}$ | $0.584_{027}$ | $0.307_{052}$ | $0.361$ |
| | w/ EGNN | $0.450_{041}$ | $0.416_{060}$ | $0.193_{153}$ | $0.566_{098}$ | $0.342_{020}$ | $0.118_{094}$ | $0.091_{065}$ | $0.095_{027}$ | $0.558_{045}$ | $0.307_{049}$ | $0.313$ |
| | w/ E2CNN | $0.450_{042}$ | $0.301_{027}$ | $0.440_{046}$ | $0.574_{049}$ | $0.337_{022}$ | $0.121_{079}$ | $0.270_{078}$ | $0.059_{020}$ | $0.504_{005}$ | $0.293_{047}$ | $0.334$ |
| Gigapath | STFlow | $0.565_{055}$ | $0.415_{013}$ | $0.513_{063}$ | $0.651_{089}$ | $0.325_{023}$ | $0.260_{023}$ | $0.326_{065}$ | $0.125_{019}$ | $0.602_{013}$ | $0.305_{053}$ | $0.409$ |
| | w/ EGNN | $0.565_{067}$ | $0.410_{012}$ | $0.505_{054}$ | $0.602_{069}$ | $0.325_{021}$ | $0.233_{046}$ | $0.295_{043}$ | $0.106_{014}$ | $0.586_{018}$ | $0.294_{066}$ | $0.392$ |
| | w/ E2CNN | $0.544_{068}$ | $0.376_{013}$ | $0.470_{052}$ | $0.623_{042}$ | $0.304_{002}$ | $0.225_{055}$ | $0.294_{098}$ | $0.102_{007}$ | $0.549_{022}$ | $0.271_{051}$ | $0.375$ |
| UNI | STFlow | $0.589_{063}$ | $0.420_{005}$ | $0.506_{078}$ | $0.707_{028}$ | $0.328_{013}$ | $0.243_{002}$ | $0.335_{070}$ | $0.128_{017}$ | $0.608_{021}$ | $0.305_{056}$ | $0.419$ |
| | w/ EGNN | $0.578_{069}$ | $0.410_{002}$ | $0.495_{076}$ | $0.662_{028}$ | $0.321_{041}$ | $0.239_{016}$ | $0.319_{052}$ | $0.109_{026}$ | $0.590_{019}$ | $0.290_{051}$ | $0.401$ |
| | w/ E2CNN | $0.562_{077}$ | $0.236_{091}$ | $0.454_{065}$ | $0.670_{033}$ | $0.327_{007}$ | $0.220_{041}$ | $0.302_{140}$ | $0.094_{009}$ | $0.498_{032}$ | $0.263_{055}$ | $0.362$ |

**Ablation study** In this experiment, we perform an ablation study to evaluate the impact of STFlow's core modules. Specifically, we individually disable the flow matching learning framework (w/o FM) and the frame averaging-related transformations (w/o FA). The experimental results are present in Table 4. As shown in the table, we can observe that removing any of the core modules leads to performance degradation to varying degrees, and this trend remains consistent across different pathology foundation models.

Table 4: Ablation study.

| | | IDC | PRAD | PAAD | SKCM | COAD | READ | CCRCC | HCC | LUNG | LYMPH | Avg. |
|---|---|---|---|---|---|---|---|---|---|---|---|---|
| Ciga | STFlow | $0.460_{028}$ | $0.380_{001}$ | $0.440_{047}$ | $0.608_{072}$ | $0.344_{023}$ | $0.137_{075}$ | $0.250_{054}$ | $0.105_{030}$ | $0.584_{027}$ | $0.307_{052}$ | $0.361$ |
| | w/o FM | $0.436_{021}$ | $0.380_{003}$ | $0.419_{040}$ | $0.593_{054}$ | $0.336_{028}$ | $0.126_{107}$ | $0.240_{043}$ | $0.095_{040}$ | $0.585_{025}$ | $0.296_{051}$ | $0.350$ |
| | w/o FA | $0.450_{040}$ | $0.375_{003}$ | $0.436_{074}$ | $0.580_{092}$ | $0.323_{017}$ | $0.125_{097}$ | $0.239_{060}$ | $0.093_{034}$ | $0.579_{030}$ | $0.290_{054}$ | $0.349$ |
| Gigapath | STFlow | $0.565_{055}$ | $0.415_{013}$ | $0.513_{063}$ | $0.651_{089}$ | $0.325_{023}$ | $0.260_{023}$ | $0.326_{065}$ | $0.125_{019}$ | $0.602_{013}$ | $0.305_{053}$ | $0.409$ |
| | w/o FM | $0.563_{069}$ | $0.411_{004}$ | $0.506_{063}$ | $0.650_{065}$ | $0.300_{028}$ | $0.228_{046}$ | $0.325_{064}$ | $0.117_{014}$ | $0.598_{009}$ | $0.281_{058}$ | $0.398$ |
| | w/o FA | $0.560_{056}$ | $0.418_{007}$ | $0.506_{066}$ | $0.616_{090}$ | $0.328_{005}$ | $0.251_{028}$ | $0.301_{053}$ | $0.112_{020}$ | $0.592_{014}$ | $0.290_{053}$ | $0.397$ |
| UNI | STFlow | $0.589_{063}$ | $0.420_{005}$ | $0.506_{078}$ | $0.707_{028}$ | $0.328_{013}$ | $0.243_{002}$ | $0.335_{070}$ | $0.128_{017}$ | $0.608_{021}$ | $0.305_{056}$ | $0.419$ |
| | w/o FM | $0.580_{065}$ | $0.420_{008}$ | $0.488_{080}$ | $0.705_{039}$ | $0.316_{028}$ | $0.235_{029}$ | $0.322_{041}$ | $0.116_{028}$ | $0.606_{010}$ | $0.277_{057}$ | $0.404$ |
| | w/o FA | $0.583_{059}$ | $0.419_{011}$ | $0.500_{083}$ | $0.670_{046}$ | $0.322_{032}$ | $0.240_{005}$ | $0.307_{036}$ | $0.111_{024}$ | $0.599_{018}$ | $0.300_{058}$ | $0.405$ |

## 4.3 BIOMARKER DISCOVERY

One of the important applications of spatial gene expression prediction is to understand disease progression in relation to tissue morphology. In this section, we present a case study on two invasive ductal carcinoma (IDC) samples imaged with Xenium. We visualize the expression levels of two genes: GATA3 and ERBB2, which are both known prognostic markers in breast cancer (Mehra et al., 2005; Revillion et al., 1998). For a clear visualization, the ground-truth and predicted gene expression levels are normalized, as shown in Figure 3.

The results demonstrate a strong correlation between STFlow's predictions and the ground-truth gene expression. For instance, compared to the state-of-the-art baseline TRIPLEX on sample TENX95, STFlow achieves a correlation of 0.891 vs 0.86 for GATA3 and 0.913 vs 0.887 for ERBB2. Based on

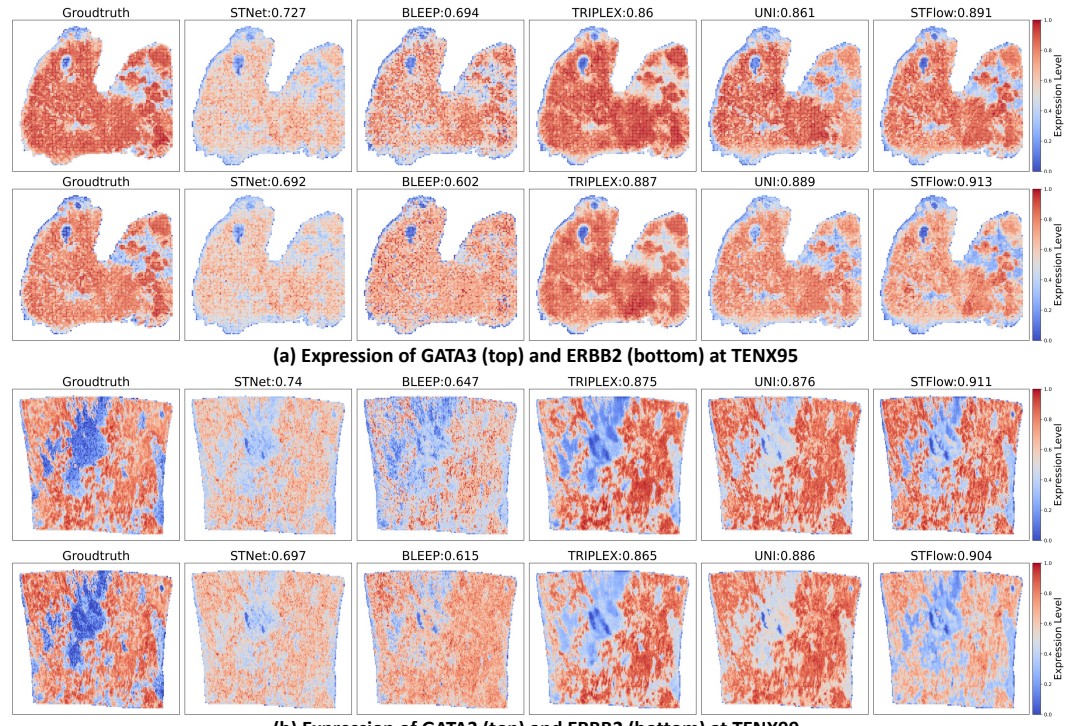

Figure 3: STFlow for Biomarker Discovery in Breast Samples: **(a)** TENX95 and **(b)** TENX99. The top row of subfigures shows gene GATA3, while the bottom row shows gene ERBB2. The Pearson correlation between ground truth and predictions is provided in each subfigure's title.

the heatmap visualizations, we can observe a great alignment of STFlow's predictions with the ground-truth gene expression patterns. Another interesting observation is that building on top of the visual features extracted by UNI, STFlow achieves a higher correlation due to its more accurate prediction of low expression levels, i.e., the blue area shown in the figures. We attribute this improvement to the iterative refinement process which can progressively adjust the predictions, better capturing subtle gene expression patterns.

## 5  CONCLUSION

In this paper, we study the problem of gene expression prediction from histology images. Despite the promising results achieved by the previous methods, we argue that gene interaction which is a key factor regulating gene expression has been overlooked. Motivated by this, we propose STFlow, a flow matching framework incorporating gene-gene dependency with an iterative refinement paradigm. The zero-inflated negative binomial distribution is applied as the prior distribution for utilizing the inductive bias of the gene expression data. Specifically, the denoiser architecture is a frame-averaging Transformer that integrates spatial context and gene interactions within the attention mechanism. Our experimental results across 10 benchmarks show that STFlow consistently outperforms the SOTA baseline methods.

**Limitation**  Our learning framework does not currently include the estimation of the hyperparameters for the ZINB distribution; instead, we use a grid search to identify the optimal hyperparameter combination. A potential improvement would be to initially employ the empirical distribution or a distribution estimation model, such as a Variational Autoencoder (VAE), to estimate the ZINB hyperparameters based on the training set.

**Reproducibility**  The implementation details, including hyperparameters and the GitHub repositories for each method, are provided in Appendix A. Additionally, the implementation of STFlow and the

experimental pipelines are available in an anonymous repository, linked in the footnote on the first page.

**Ethics Statement** This paper presents work whose goal is to advance the field of spatial transcriptomics prediction on histology images. All the datasets used in this study are publicly available. There are some potential societal consequences of our work, none of which we feel must be specifically highlighted here.

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

# A    IMPLEMENTATION

**Running environment**    The experiments are conducted on a single Linux server with The AMD EPYC 7763 64-Core Processor, 1024G RAM, and 8 RTX A6000-48GB. Our method is implemented on PyTorch 2.3.0 and Python 3.10.14.

**Training details**    For all the models, we fix the optimizer as Adam (Kingma & Ba, 2014) and MSE loss as the loss function. The gradient norm is clipped to 1.0 in each training step to ensure learning stability. The learning rate is tuned within {1e-3, 5e-4, 1e-4} and is set to 5e-4 by default, as it generally yields the best performance. Following HEST-1k, all performance metrics are reported using a cross-validation setup, with the mean and standard deviation calculated across the different splits. Besides, all the weights of pathology foundation models are frozen.

For each model, we search the hyperparameters in the following ranges: the dropout rate in {0, 0.2, 0.5}, the number of nearest neighbors for the slide-based methods in {4, 8, 25}, and the number of attention heads in {1, 2, 4, 8}. All models are trained for 100 epochs, with early stopping applied if no performance improvement is observed for 20 epochs. The implementation and hyperparameters used in each method are shown below:

- STFlow: The number of layers, attention heads, and neighbors are 4, 4, and 8, respectively. Besides, dropout and hidden sizes are set at 0.2 and 128. The number of sampling steps for flow matching is set as 5. For the ZINB distribution, zero-inflation probability is fixed as 0.5, the mean is searched {0.1,0.2,0.4}, and the number of failures is searched in {1,2,4}. For efficient training, each sample is a randomly selected continuous region from the WSI, with its size determined by a proportion sampled from a uniform distribution ranging from 0 to 1. In each training step, we will sample a region from WSI.

- Ciga[4], UNI[5], and Gigapath[6]: We download the pretrained weight from the official repository and normalize the input images using the ImageNet mean and standard deviation. The Random Forest model with 70 trees serves as the linear head. Additionally, Gigapath offers three different pretrained versions; we selected the one with the largest hidden size, i.e., "gigapath_slide_enc12l1536d". For the Gigapath-slide, all the spot images of a WSI are input for global attention.

- STNet[7]: Following the official implementation, we use a pretrained DenseNet121 as the image encoder and an MLP as the linear head. The input spot images are randomly augmented with horizontal flips and rotations and are then normalized using the ImageNet mean and standard deviation. The batch size, i.e., the number of spot images in each training step, is 128.

- BLEEP[8]: This method trains an image encoder and a gene expression encoder using contrastive loss. For a given spot image, it retrieves the gene expressions of similar spots from a reference set, using the average expression of these spots as the prediction. We use a pretrained ResNet50 as the image encoder and MLPs as linear heads to project the extracted visual features and gene expressions. The temperature for the contrastive loss is set to 1, and the number of retrieved spots is 50. For a fair comparison, we directly use the training WSI as the reference set, as there are no additional splits in HEST-1k. The batch size is set as 128.

- Hist2ST[9]: The architecture of Hist2ST includes a convolution network, a Transformer, and a GNN. The coordinates are embedded with a linear layer. The final representation is aggregated across each GNN layer's output with an LSTM. The number of layers for each model is 2, 4, and 8. The input spot images are randomly augmented with horizontal flips and rotations and are then normalized using the ImageNet mean and standard deviation. Similar to STFlow, each training sample is a sampled region of WSI.

- HisToGene[10]: This model includes a ViT for encoding spot images within the WSI. The number of layers, the number of attention heads, the dropout rate, and the hidden size are set as 4, 16, 0.1,

---

[4] https://github.com/ozanciga/self-supervised-histopathology
[5] https://huggingface.co/MahmoodLab/UNI
[6] https://huggingface.co/prov-gigapath/prov-gigapath
[7] https://github.com/bryanhe/ST-Net/tree/master
[8] https://github.com/bowang-lab/BLEEP/tree/main
[9] https://github.com/biomed-AI/Hist2ST/tree/main
[10] https://github.com/maxpmx/HisToGene

and 128. The coordinates are embedded with a linear layer. The input spot images are randomly augmented with horizontal flips and rotations and are then normalized using the ImageNet mean and standard deviation. For efficient training, we sample a continuous region from WSI in each training step, similar to STFlow.

- TRIPLEX[11]: This model comprises a target encoder for the target spot, a local encoder for the neighboring spots, a global encoder for WSI, and a fusion encoder for combining all these representations. In line with the official implementation, the spot images are first embedded using Ciga before being fed into the model. Each encoder is configured with 2 layers, 8 attention heads, and a dropout rate of 0.1. The local encoder considers 25 neighboring spots. Additionally, the coordinates are embedded using a proposed atypical position encoding generator based on a convolutional network. A continuous region from WSI is sampled for each training step, using the same strategy as STFlow.

Here we also provide the implementation of the E(2)-invariant encoder baselines:

- EGNN[12]: Similar to a standard GNN, EGNN propagates representations from neighboring spots to the target spots and uses MLPs for transformation, incorporating the distances between them in the calculations. The number of layers and neighbors is set to 4 and 8, respectively, with a hidden size of 128 and a dropout rate of 0.2. For a fair comparison, EGNN leverages the visual features extracted by the pathology foundation model and is integrated with the flow matching framework.

- E2CNN[13]: E2CNN is an $E(n)$ convolution framework that implements various equivariant operations, such as convolution layers, batchnorm, and pooling layers. Here, we use the 10-layer ResNet from the official codebase as the backbone. To construct the input batch, each spot and its surrounding neighbors are arranged into a $5 \times 5$ grid with the target spot at the center. The visual features extracted by the foundation models are then stacked as channels, resulting in a tensor of dimensions $d \times 5 \times 5$.

## B  RUNNING TIME COMPARISON

To demonstrate the efficiency of STFlow, we present the average inference time on the test set of each dataset across splits, as shown in Table 5. Since STFlow does not require training an image encoder, we separate the time spent on the pathology foundation model from the multi-step denoising (STFlow w/o $f_{\mathrm{pfm}}$) for a fair comparison. The reported times for other methods include the time required for image encoding.

Notably, our proposed architecture is highly efficient due to its use of local neighbors, but the primary inference bottleneck lies in the pathology foundation models. This bottleneck could be alleviated through acceleration techniques, such as mixed precision inference and model quantization.

Table 5: Inference time comparison.

|  | IDC | PRAD | PAAD | SKCM | COAD | READ | CCRCC | HCC | LUNG | LYMPH |
|---|---|---|---|---|---|---|---|---|---|---|
| STNet | 29.61s | 87.12s | 6.11s | 2.37s | 43.82s | 5.46s | 18.13s | 10.33s | 6.53s | 15.00s |
| BLEEP | 27.72s | 112.43s | 7.78s | 10.77s | 14.76s | 27.78s | 220.06s | 3.79s | 4.69s | 15.46s |
| Hist2ST | 16.41s | 110.01s | 6.89s | 3.01s | 10.80s | 15.41s | 30.04s | 2.86s | 7.65s | 14.14s |
| HisToGene | 12.03s | 95.91s | 12.85s | 5.46s | 12.30s | 13.04s | 36.93s | 2.37s | 4.19s | 11.01s |
| TRIPLEX | 39.67s | 131.99s | 6.80s | 2.88s | 44.17s | 11.07s | 42.26s | 10.09s | 11.35s | 16.19s |
| STFlow w/o $f_{\mathrm{pfm}}$ | 0.51s | 1.50s | 0.16s | 0.14s | 0.41s | 0.26s | 0.64s | 0.15s | 0.78s | 0.25s |
| Ciga | 10.27s | 38.20s | 3.02s | 1.95s | 9.11s | 5.03s | 14.91s | 2.45s | 3.17s | 5.64s |
| UNI | 64.35s | 237.87s | 20.59s | 9.77s | 56.47s | 30.01s | 92.49s | 12.98s | 17.37s | 33.04s |
| Gigapath | 233.09s | 867.29s | 50.13s | 30.62s | 203.92s | 106.13s | 336.47s | 44.63s | 59.69s | 117.25s |

---

[11]https://github.com/NEXGEM/TRIPLEX/tree/main

[12]https://github.com/vgsatorras/egnn

[13]https://github.com/QUVA-Lab/e2cnn/tree/master

# C  DATASET

Table 6 lists the statistics of the benchmark datasets. Further details about these datasets can be found in Jaume et al. (2024). Note that the COAD dataset differs from the version in Jaume et al. (2024), as it was updated two month after the paper's release.

Table 6: Dataset statistics.

|  | IDC | PRAD | PAAD | SKCM | COAD | READ | CCRCC | HCC | LUNG | LYMPH |
|---|---|---|---|---|---|---|---|---|---|---|
| Organ | Breast | Prostate | Pancreas | Skin | Colon | Rectum | Kidney | Liver | Lung | Axillary Lymph Nodes |
| Technology | Xenium | Visium | Xenium | Xenium | Visium | Visium | Visium | Visium | Xenium | Visium |
| #Patients | 4 | 2 | 3 | 2 | 3 | 2 | 24 | 2 | 2 | 4 |
| #Samples | 4 | 23 | 3 | 2 | 6 | 4 | 24 | 2 | 2 | 4 |
| #Splits | 4 | 2 | 3 | 2 | 2 | 2 | 6 | 2 | 2 | 4 |
| Avg. spots | 4925 | 2454 | 2780 | 1741 | 5079 | 1909 | 2792 | 1941 | 1944 | 4990 |

