# OpenReview forum: "Predicting Spatial Transcriptomics from Histology Images via Biologically Informed Flow Matching"
_ICLR.cc/2025/Conference — Submitted to ICLR 2025_

### Official Review · Reviewer_2qgH · 2024-11-03

**Soundness:** 3
**Presentation:** 3
**Contribution:** 2
**Rating:** 5
**Confidence:** 4

**Summary:**

The paper proposes a method 'STFlow' to predict gene expression from H&E images for the Spots in ST-Data. The proposed method uses flow matching starting from data sampled from zero-inflated negative binomial distribution to account for the sparsity in the gene data. The model is a VIT based architecture and the attention is constructed to take into account K nearest neighbors spots predicted gene expressions at time T, their relative positions, and their image encodings. The authors evaluate their  method on benchmarks from the Hest1K paper and compare with spot based and slide based approaches using different backbones. Using STFlow with different foundational models shows improved performance to other baselines. The paper also includes ablation studies for different prior distributions and E(2) representation invariance approaches.

**Strengths:**

- The paper is well written and easy to follow.
- The proposed method seems effective for the application and surpasses all other baselines.
- The authors provide all the hyperparameters for their experiments.

**Weaknesses:**

- The novelty in the method is limited / incremental.

- The motivation for ZINB priors is not clear given the ablation study showing that using zero priors gives almost same results.
In the ablation studies, using "zero distribution, where all samples are zero". I'm not sure what it means to apply Log1p to the samples. That would mean Log 0 which is zero. This needs to be clarified. Also, the results with the zero distribution see very close to ZINB. So why go through the trouble of esimtating the parameters for ZINB? and they are better than Gaussian which sounds counterintuitive but is not explained.

- How does the proposed method compare to recent methods using diffusion models for the same task, such as: stDiff: a diffusion model for imputing spatial transcriptomics through single-cell transcriptomics, Briefings in Bioinformatics, Volume 25, Issue 3, May 2024.

- Results with ResNet50 would be insightful as to how much power is obtained from the image encoding. Similarily, a model like BLEEP is using a ResNet50 but using embeddings from more recent foundational models would make a more fair comparison to the method.

- The qualitative results in Fig 3 (a) are not convincing, specially when we visually compare the Triplex results to STFLow on TENX95. Even though the reported numbers for STFlow show higher correlation, TRIPLEX results look better.

- In Table 2, Row 4 is a copy of row 7. Looks like it was copied by mistake.

**Questions:**

Please refer to the weaknesses

---

> ### Author Response · Authors · 2024-11-25
>
> ```
> Weakness: The novelty in the method is limited / incremental.
> ```
>
> Respond: We respectfully disagree with the statements and would like to emphasize the contributions of our paper. As for the motivation, previous slide-level studies have primarily focused on designing better encoders to capture the global context, such as Hist2ST and TRIPLEX. However, they overlooked two critical aspects: **(1)** leveraging the representational power of pathology foundation models to improve prediction accuracy, and **(2)** explicitly incorporating gene-gene dependencies to infer gene expression. Inspired by these gaps, STFlow builds on top of pathology foundation models and introduces a generative approach to learning the landscape of gene regulation. This enables more accurate predictions. I believe our work will inspire the development of generative models in this domain.
>
> As for the technical contributions, we do not consider the introduced flow matching to be trivial:
>
> - We use the ZINB distribution to model the overdispersion and sparsity of gene expression effectively.
> - Our proposed spatial attention mechanism is specifically designed for flow matching. The E(2)-invariant attention integrates gene expression to regulate the attention map, akin to the way cells regulate each other.
>
> ```
> Weakness: The motivation for ZINB priors is not clear given the ablation study showing that using zero priors gives almost the same results. In the ablation studies, using "zero distribution, where all samples are zero". I'm not sure what it means to apply Log1p to the samples. That would mean Log 0 which is zero. This needs to be clarified. Also, the results with the zero distribution see very close to ZINB. So why go through the trouble of esimtating the parameters for ZINB? and they are better than Gaussian which sounds counterintuitive but is not explained.
> ```
>
> Respond: We apologize for any confusion. Zero distribution is indeed all zero distribution so Log1p will produce the same results. This statement is removed from our updated manuscript.
>
> As for the motivation for using ZINB, gene expression in ST data exhibits a similar pattern to scRNA-seq data:
>
> 1. **Non-activated genes dominate**: Most of the data consists of non-activated genes with expression values equal to 0.
> 2. **Over-dispersion**: Gene expression variance exceeds the mean value.
>
> Below are the statistics for HEST-1k. Here, **Exp=0(%)** represents the average proportion of zero-expression values in the gene expression data across samples, while **mean/variance** indicates the average mean and variance of gene expression across samples.
>
> |  | Exp=0(%) | Mean | Variance |
> | --- | --- | --- | --- |
> | IDC | 0.588 | 12.307 | 2829.614 |
> | PRAD | 0.939 | 0.194 | 21.406 |
> | PAAD | 0.561 | 10.807 | 3357.701 |
> | SKCM | 0.702 | 13.652 | 6010.685 |
> | COAD | 0.690 | 9.144 | 3541.19 |
> | READ | 0.887 | 0.336 | 15.52 |
> | CCRCC | 0.845 | 0.426 | 34.242 |
> | HCC | 0.855 | 0.886 | 276.34 |
> | LUNG | 0.509 | 12.084 | 2673.61 |
> | LYMPH_IDC | 0.837 | 0.605 | 96.083 |
>
> Such an observation motivates us to use the ZINB distribution, where the negative binomial component explicitly models dispersion with an additional parameter to adjust the variance, and the zero-inflated condition captures the sparsity in gene expression.
>
> In terms of performance, we would like to clarify that comparing the average Pearson correlation across datasets might somewhat **underestimate the contribution of flow matching** since there are significant scale differences between datasets (e.g., the model achieves 0.707 on SKCM but only 0.128 on HCC). Here, we present the relative improvement of the model with flow matching on each dataset, where flow matching can provide around 5% improvement on UNI, which is non-trivial.
>
> | relative improvement% | Average | IDC | PRAD | PAAD | SKCM | COAD | READ | CCRCC | HCC | LUNG | LYMPH_IDC |
> | --- | --- | --- | --- | --- | --- | --- | --- | --- | --- | --- | --- |
> | UNI | 5.04% | 0.68% | 5.79% | 2.42% | 3.06% | 2.01% | 3.84% | 3.39% | 24.27% | 0% | 4.81% |
> | Gigapath | 1.37% | 0.17% | 0.97% | 1.38% | 0% | 0.61% | -0.38% | -0.60% | 8.69% | 1.51% | 1.32% |
>
> We will incorporate these analyses in our manuscript.

---

> > ### Author Response · Authors · 2024-11-25
> >
> > ```
> > Weakness: How does the proposed method compare to recent methods using diffusion models for the same task, such as: stDiff: a diffusion model for imputing spatial transcriptomics through single-cell transcriptomics, Briefings in Bioinformatics, Volume 25, Issue 3, May 2024.
> > ```
> >
> > Respond: We would like to clarify that stDiff is designed for the spatial transcriptomics **imputation** task, where the expression of certain genes is masked, and the model predicts these expressions based on the remaining gene data. In contrast, STFlow aims to predict gene expression directly from histology images without relying on any existing gene expression information. We will add this discussion to our related work.
> >
> > ```
> > Weakness: Results with ResNet50 would be insightful as to how much power is obtained from the image encoding. Similarily, a model like BLEEP is using a ResNet50 but using embeddings from more recent foundational models would make a more fair comparison to the method.
> > ```
> >
> > Respond: The results of BLEEP and TRIPLEX using Ciga and UNI are shown below:
> >
> > |  | Ciga |  |  | UNI |  |  |
> > | --- | --- | --- | --- | --- | --- | --- |
> > |  | BLEEP | TRIPLEX | STFlow | BLEEP | TRIPLEX | STFlow |
> > | IDC | 0.436(.042) | 0.492(.042) | 0.460(.028) | 0.521(.068) | 0.609(.079) | 0.589(.063) |
> > | PRAD | 0.361(.004) | 0.351(.023) | 0.380(.001) | 0.375(.010) | 0.385(.004) | 0.420(.005) |
> > | PAAD | 0.411(.060) | 0.429(.045) | 0.440(.047) | 0.463(.065) | 0.489(.068) | 0.506(.078) |
> > | SKCM | 0.550(.062) | 0.576(.091) | 0.608(.072) | 0.583(.008) | 0.698(.023) | 0.707(.028) |
> > | COAD | 0.289(.009) | 0.305(.004) | 0.344(.023) | 0.293(.024) | 0.319(.033) | 0.328(.013) |
> > | READ | 0.137(.040) | 0.129(.062) | 0.137(.075) | 0.236(.014) | 0.213(.066) | 0.243(.002) |
> > | CCRCC | 0.221(.052) | 0.229(.036) | 0.250(.054) | 0.284(.068) | 0.302(.039) | 0.335(.070) |
> > | HCC | 0.066(.034) | 0.044(.022) | 0.105(.030) | 0.094(.021) | 0.076(.031) | 0.128(.017) |
> > | LUNG | 0.531(.008) | 0.563(.036) | 0.584(.027) | 0.584(.012) | 0.602(.028) | 0.608(.021) |
> > | LYMPH_IDC | 0.250(.050) | 0.286(.055) | 0.307(.052) | 0.228(.055) | 0.294(.056) | 0.305(.056) |
> > | Average | 0.325 | 0.340 | 0.361 | 0.366 | 0.398 | 0.419 |
> >
> > Notably, STNet simply trains a linear head on top of the image encoder, so replacing its image encoder is equivalent to the case of UNI and Ciga with a linear head. Therefore, we only present the performance of BLEEP and TRIPLEX here.
> >
> > All these results will be included in our updated manuscript.
> >
> > ```
> > Weakness: The qualitative results in Fig 3 (a) are not convincing, specially when we visually compare the Triplex results to STFLow on TENX95. Even though the reported numbers for STFlow show higher correlation, TRIPLEX results look better.
> > ```
> >
> > Respond: In the case of TENX95, TRIPLEX tends to produce higher overall predictions (with a general reddish bias), which can skew its accuracy in predicting low-expression regions. This bias makes it less effective at capturing the sparsity patterns in the data, particularly in low-expressed areas. In contrast, STFlow can more accurately predict the low-expressed region (blue region), which contributes to the higher correlation. Such an improvement is attributed to the introduced prior, i.e., ZINB distribution, which explicitly models the sparsity in gene expression through a dropout probability.
> >
> > ```
> > Weakness: In Table 2, Row 4 is a copy of row 7. Looks like it was copied by mistake.
> > ```
> >
> > Respond: We apologize for any confusion. The results are updated in the manuscript.

---

> > > ### Comment · Reviewer_2qgH · 2024-12-02
> > >
> > > I thank the authors for addressing my concerns. I am raising my score, however, I still find the novelty limited.

---

> > > > ### Author Response · Authors · 2024-12-02
> > > >
> > > > Thank you for recognizing our efforts! We hope our clarification on the novelty can address your concerns. Please feel free to let us know if you have any further questions.

---

### Official Review · Reviewer_83ta · 2024-11-04

**Soundness:** 3
**Presentation:** 3
**Contribution:** 3
**Rating:** 6
**Confidence:** 4

**Summary:**

This paper introduces STFlow, using history imaging to predict spatial transcriptomics (ST). STFlow is a flow-based generative model to predict ST from whole slide image. STFlow chooses a zero-inflated negative binomial distribution as prior distribution. STFlow models the interaction of of gene across different spots compared to previous methods that predict at each spot independently. On a HEST1K benchmark, STFlow outperforms all baselines.

**Strengths:**

This paper focus on an important steps of predicting ST from WSI. It uses an innovative flow-based generative model. By incorporating spatial attention, STFlow captures dependencies between neighboring spots, reflecting the biological reality of gene regulatory networks.

**Weaknesses:**

Motivation of using ZINB prior is not strong: Is there a specific reason of using this distribution? Cited literatures are single cell RNA seq which is not ST. In the Table 2, it's also clear that ZINB is not helping especially for UNI and gigapath.

Notation is not easy to follow: It's claiming that algorithm 2 has a sampling factor which is gradually decrease. However, the last step is basically, $Y_{t+1} \leftarrow Y_t + (\hat{Y} - Y_t) / (T-t)$, I don't think it's decreasing?

Different range of t in training and inference. It's clear the t in train is from 0 ~ 1 which in inference is from 0 to T - 1. I don't know how to model account for different range of t.

Limited Generalization Under Transformations: The paper mentions that the pathology foundation models used are not E(2)-invariant, potentially restricting the model's ability to generalize under certain spatial transformations. This could limit the applicability of STFlow in diverse datasets with varying orientations and scales.

Limited Dataset Diversity: The evaluation primarily focuses on the HEST-1k benchmark. Including additional datasets like STImage-1K4M from varied sources could strengthen the claims of generalizability and robustness.

**Questions:**

What's the motivation of using ZINB? How does this generalize to ST? Is there any other alternative?

Could you use a more clear way of presenting algorithm 2?

Please address the issue of different range of t.

Exploring or integrating E(2)-invariant architectures for the pathology foundation models could enhance the overall invariance of STFlow, improving its robustness to spatial transformations.

Please consider more dataset like STImage-1K4M.

---

> ### Author Response · Authors · 2024-11-25
>
> ```
> Weakness: Motivation of using ZINB prior is not strong: Is there a specific reason of using this distribution? Cited literatures are single cell RNA seq which is not ST. In the Table 2, it's also clear that ZINB is not helping especially for UNI and gigapath.
> ```
>
> Respond: Gene expression in ST data exhibits a similar pattern to scRNA-seq data:
>
> 1. **Non-activated genes dominate**: Most of the data consists of non-activated genes with expression values equal to 0.
> 2. **Over-dispersion**: Gene expression variance exceeds the mean value.
>
> Below are the statistics for HEST-1k. Here, **Exp=0(%)** represents the average proportion of zero-expression values in the gene expression data across samples, while **mean/variance** indicates the average mean and variance of gene expression across samples.
>
> |  | Exp=0(%) | Mean | Variance |
> | --- | --- | --- | --- |
> | IDC | 0.588 | 12.307 | 2829.614 |
> | PRAD | 0.939 | 0.194 | 21.406 |
> | PAAD | 0.561 | 10.807 | 3357.701 |
> | SKCM | 0.702 | 13.652 | 6010.685 |
> | COAD | 0.690 | 9.144 | 3541.19 |
> | READ | 0.887 | 0.336 | 15.52 |
> | CCRCC | 0.845 | 0.426 | 34.242 |
> | HCC | 0.855 | 0.886 | 276.34 |
> | LUNG | 0.509 | 12.084 | 2673.61 |
> | LYMPH_IDC | 0.837 | 0.605 | 96.083 |
>
> Such an observation motivates us to use the ZINB distribution, where the negative binomial component explicitly models dispersion with an additional parameter to adjust the variance, and the zero-inflated condition captures the sparsity in gene expression.
>
> In terms of performance, we would like to clarify that comparing the average Pearson correlation across datasets might somewhat **underestimate the contribution of flow matching** since there are significant scale differences between datasets (e.g., the model achieves 0.707 on SKCM but only 0.128 on HCC). Here, we present the relative improvement of the model with flow matching on each dataset, where flow matching can provide around 5% improvement on UNI, which is non-trivial.
>
> | relative improvement% | Average | IDC | PRAD | PAAD | SKCM | COAD | READ | CCRCC | HCC | LUNG | LYMPH_IDC |
> | --- | --- | --- | --- | --- | --- | --- | --- | --- | --- | --- | --- |
> | UNI | 5.04% | 0.68% | 5.79% | 2.42% | 3.06% | 2.01% | 3.84% | 3.39% | 24.27% | 0% | 4.81% |
> | Gigapath | 1.37% | 0.17% | 0.97% | 1.38% | 0% | 0.61% | -0.38% | -0.60% | 8.69% | 1.51% | 1.32% |
>
> We will incorporate these analyses in our manuscript.
>
> ```
> Weakness: Notation is not easy to follow: It's claiming that algorithm 2 has a sampling factor which is gradually decrease. However, the last step is basically, Yt+1←Yt+(Y^−Yt)/(T−t), I don't think it's decreasing?
> ```
>
> Respond: We apologize for any confusion. The coefficient indeed increases as the time step progresses, ensuring the prediction gradually converges to the final result. The prediction at the final step is directly returned (refer to Line 6 in Algorithm 2). The description has been updated in the revised manuscript, highlighted in red.

---

> > ### Author Response · Authors · 2024-11-25
> >
> > ```
> > Weakness: Different range of t in training and inference. It's clear the t in train is from 0 ~ 1 which in inference is from 0 to T - 1. I don't know how to model account for different range of t.
> > ```
> >
> > Respond: We apologize for any confusion. The notation has been corrected in the updated manuscript, where $s$ is now used to denote the refinement step.
> >
> > ```
> > Weakness: Limited Generalization Under Transformations: The paper mentions that the pathology foundation models used are not E(2)-invariant, potentially restricting the model's ability to generalize under certain spatial transformations. This could limit the applicability of STFlow in diverse datasets with varying orientations and scales.
> > ```
> >
> > Respond: The pathology foundation models are pretrained with extensive image augmentations, including rotation, translation, and reflection, making the extracted spot features robust to any E(2) transformation. This robustness is demonstrated by the strong performance of STFlow across 10 HEST benchmark datasets with different technologies.
> >
> > Additionally, we evaluate the performance of STFlow under random spot-level transformations. Specifically, using a trained STFlow model on the original training set, we compare its performance on the test set with and without applying random E(2) transformations to the spot images:
> >
> > |  | STFlow | STFlow w/ E(2) transformation on spots |
> > | --- | --- | --- |
> > | IDC | 0.589(.063) | 0.586(.068) |
> > | PRAD | 0.420(.005) | 0.419(.002) |
> > | PAAD | 0.506(.078) | 0.508(.080) |
> > | SKCM | 0.707(.028) | 0.709(.053) |
> > | COAD | 0.328(.013) | 0.331(.027) |
> > | READ | 0.243(.002) | 0.244(.028) |
> > | CCRCC | 0.335(.070) | 0.334(.056) |
> > | HCC | 0.128(.017) | 0.125(.010) |
> > | LUNG | 0.608(.021) | 0.607(.013) |
> > | LYMPH_IDC | 0.305(.056) | 0.304(.061) |
> > | Avg. | 0.419 | 0.418 |
> >
> > It can be found that even without image augmentation during training, the performance can still be stable during inference.
> >
> > ```
> > Weakness: Limited Dataset Diversity: The evaluation primarily focuses on the HEST-1k benchmark. Including additional datasets like STImage-1K4M from varied sources could strengthen the claims of generalizability and robustness.
> > ```
> >
> > Respond: We here present the comparison results using STImage-1K4M. Similar to HEST-1K, we create benchmarks by selecting cancer samples for each organ and randomly splitting the train/validation/test sets (8:1:1) at the slide level. The statistics are shown below:
> >
> > |  | breast | brain | skin | mouth | stomach | prostate | colon |
> > | --- | --- | --- | --- | --- | --- | --- | --- |
> > | #samples | 189 | 152 | 16 | 16 | 12 | 7 | 4 |
> >
> > The model predicts the top 50 highly variable genes, with Pearson correlation used as the evaluation metric. Note that we only include organs where the models achieve significant correlations (at least >0.1). All results are reported based on 5 different random seeds, and all models use UNI as the image encoder.
> >
> > |  | UNI | BLEEP | TRIPLEX | STFlow |
> > | --- | --- | --- | --- | --- |
> > | breast | 0.395(.000) | 0.244(.040) | 0.466(.010) | 0.428(.025) |
> > | brain | 0.343(.001) | 0.299(.008) | 0.338(.005) | 0.367(.021) |
> > | skin | 0.137(.001) | 0.160(.023) | 0.163(.005) | 0.133(.022) |
> > | mouth | 0.173(.000) | 0.187(.030) | 0.121(.005) | 0.162(.018) |
> > | prostate | 0.217(.000) | 0.271(.024) | 0.267(.042) | 0.313(.010) |
> > | stomach | 0.164(.003) | 0.188(.033) | 0.192(.024) | 0.283(.029) |
> > | colon | 0.276(.003) | 0.301(.015) | 0.243(.003) | 0.288(.000) |
> > | Avg. | 0.243 | 0.235 | 0.255 | 0.282 |
> >
> > The results demonstrate the effectiveness of STFlow, where our method can achieve over 10% improvement. The complete benchmark will be included in our updated manuscript.

---

> > > ### Author Response · Authors · 2024-12-02
> > >
> > > Thank you once again for your valuable comments on our paper! As the deadline is approaching, we may not be able to respond after that. Please let us know if you have any further questions or concerns. We are committed to addressing your feedback.

---

> > > > ### Comment · Reviewer_83ta · 2024-12-02
> > > >
> > > > I am pretty satisfied with the response and raise the score correspondingly.

---

> ### Author Response · Authors · 2024-12-02
> **Official Comment by Authors**
>
> Thank you for recognizing our efforts! Please feel free to let us know if you have any further questions.

---

### Official Review · Reviewer_qSWS · 2024-11-04

**Soundness:** 3
**Presentation:** 3
**Contribution:** 2
**Rating:** 5
**Confidence:** 4

**Summary:**

The authors presented STFlow to predict ST from histology images. STFlow uses a flow matching algorithm to predict gene expression values, and achieves a performance boost over baselines on the HEST benchmark

**Strengths:**

- Predicting spatial transcriptomics using histology data is a relevant and important problem and has a large impact on the future of computational pathology and bioinformatics research.
- The author’s approaches uses state-of-the-art ML approaches, and seems to achieves performance boost over some baselines provided.

**Weaknesses:**

- Author’s approach assembles many prior off-the-shelf methods for ST prediction, including a two-stage approach for histology, tile-level foundation models, and flow matching. Notably, the approach uses a frozen patch encoder, that does most of the heavy lifting in the representation learning, leaving it frozen inhibits the model’s ability to learn.
- The model performance boost is not substantial, and are often within error bar of the much simpler baselines. The comparison of the proposed method doesn’t have the proper slide-based baselines using the same patch encoder. For example, comparison to Hist2ST and HistToGene in table 1 does not make sense because the patch encoder is different.
- Key implementation details of the author’s approach is missing, including the model size and compute time. A comparison of the author’s model size to the baseline’s sizes provides important insight into the performance comparison.
- The authors employs leave-one-out cross validation at the patient level (which is also at the slide level for many benchmarks), except for CCRCC. Leave-one-out cross validation may lead to overfitting, and this is become more concerning here because the authors use a complex approach which can easily be overfitted on to the small number of datapoints at the slide level.

**Questions:**

Please address the concerns raised in weakness section, especially on the implementation/evaluation details and the comparison to the baselines.

---

> ### Author Response · Authors · 2024-11-25
>
> ```
> Weakness: Author’s approach assembles many prior off-the-shelf methods for ST prediction, including a two-stage approach for histology, tile-level foundation models, and flow matching. Notably, the approach uses a frozen patch encoder, that does most of the heavy lifting in the representation learning, leaving it frozen inhibits the model’s ability to learn.
> ```
>
> Respond: As mentioned in Line 159, our goal is to propose a pathology foundation model-agnostic spatial transcriptomics prediction framework, rather than to develop a better spot-level image encoder. While fine-tuning pathology image encoders with gene expression data could potentially improve performance, it is impractical, particularly for billion-level foundation models and standard gigapixel slides containing tens of thousands of image tiles. Moreover, the benchmarking is fair as all models utilize frozen image encoders.
>
> ```
> Weakness: The model performance boost is not substantial, and are often within error bar of the much simpler baselines. The comparison of the proposed method doesn’t have the proper slide-based baselines using the same patch encoder. For example, comparison to Hist2ST and HistToGene in table 1 does not make sense because the patch encoder is different.
> ```
>
> Respond: Regarding the error bar, we clarify that the variance in Table 1 represents the standard deviation of Pearson correlations **across folds within a dataset, not across random seeds**. Performance variance between folds may arise from batch effects or sample size differences. For example, a sample with few spots in the training set might lead to under-training and lower performance, while the same sample in the test set could yield better results. This pattern is also evident in UNI's performance on the READ dataset. For a clear demonstration, we here present the variance across random seeds:
>
> | STFlow (UNI) | Mean | std across folds | std across 5 random seeds |
> | --- | --- | --- | --- |
> | IDC | 0.589 | 0.063 | 0.003 |
> | PRAD | 0.420 | 0.005 | 0.002 |
> | PAAD | 0.506 | 0.078 | 0.004 |
> | SKCM | 0.707 | 0.028 | 0.005 |
> | COAD | 0.328 | 0.013 | 0.009 |
> | READ | 0.243 | 0.002 | 0.014 |
> | CCRCC | 0.335 | 0.070 | 0.003 |
> | HCC | 0.128 | 0.017 | 0.004 |
> | LUNG | 0.608 | 0.021 | 0.002 |
> | LYMPH_IDC | 0.305 | 0.056 | 0.001 |
>
> Specifically, the standard deviation across 5 random seeds is the variance in mean performance under different initializations. The results demonstrate that our model is stable across various seed initializations.
>
> As for the Hist2ST and HistToGene, these two models use a variant of ViT as their image encoder, making it unsuitable to replace their encoder with other pathology foundation models. However, we agree that the same pathology foundation models should be applied to other baselines for a comprehensive comparison. The results of BLEEP and TRIPLEX using Ciga and UNI are shown below:
>
> |  | Ciga |  |  | UNI |  |  |
> | --- | --- | --- | --- | --- | --- | --- |
> |  | BLEEP | TRIPLEX | STFlow | BLEEP | TRIPLEX | STFlow |
> | IDC | 0.436(.042) | 0.492(.042) | 0.460(.028) | 0.521(.068) | 0.609(.079) | 0.589(.063) |
> | PRAD | 0.361(.004) | 0.351(.023) | 0.380(.001) | 0.375(.010) | 0.385(.004) | 0.420(.005) |
> | PAAD | 0.411(.060) | 0.429(.045) | 0.440(.047) | 0.463(.065) | 0.489(.068) | 0.506(.078) |
> | SKCM | 0.550(.062) | 0.576(.091) | 0.608(.072) | 0.583(.008) | 0.698(.023) | 0.707(.028) |
> | COAD | 0.289(.009) | 0.305(.004) | 0.344(.023) | 0.293(.024) | 0.319(.033) | 0.328(.013) |
> | READ | 0.137(.040) | 0.129(.062) | 0.137(.075) | 0.236(.014) | 0.213(.066) | 0.243(.002) |
> | CCRCC | 0.221(.052) | 0.229(.036) | 0.250(.054) | 0.284(.068) | 0.302(.039) | 0.335(.070) |
> | HCC | 0.066(.034) | 0.044(.022) | 0.105(.030) | 0.094(.021) | 0.076(.031) | 0.128(.017) |
> | LUNG | 0.531(.008) | 0.563(.036) | 0.584(.027) | 0.584(.012) | 0.602(.028) | 0.608(.021) |
> | LYMPH_IDC | 0.250(.050) | 0.286(.055) | 0.307(.052) | 0.228(.055) | 0.294(.056) | 0.305(.056) |
> | Average | 0.325 | 0.340 | 0.361 | 0.366 | 0.398 | 0.419 |
>
> Notably, STNet simply trains a linear head on top of the image encoder, so replacing its image encoder is equivalent to the case of UNI and Ciga with a linear head. Therefore, we only present the performance of BLEEP and TRIPLEX here. All these results will be included in our updated manuscript.

---

> > ### Author Response · Authors · 2024-11-25
> >
> > ```
> > Weakness: Key implementation details of the author’s approach is missing, including the model size and compute time. A comparison of the author’s model size to the baseline’s sizes provides important insight into the performance comparison.
> > ```
> >
> > Respond: We have provided a running time comparison in Appendix B. The number of parameters are shown below:
> >
> > |  | STNet | BLEEP | Hist2ST | HistToGene | TRIPLEX | STFlow |
> > | --- | --- | --- | --- | --- | --- | --- |
> > | #parameters (M) | 0.05125 | 0.6702 | 470.6616 | 149.0469 | 13.7678 | 1.1477 |
> >
> > The results demonstrate that our model has the lowest parameter count among the slide-level baselines. Specifically, Hist2ST and HistToGene train dedicated image encoders, which introduce a significant number of parameters. TRIPLEX employs a three-branch attention mechanism, which is considerably heavier than our proposed spatial attention model.
> >
> > All these analyses will be included in our updated manuscript.
> >
> > ```
> > Weakness: The authors employs leave-one-out cross validation at the patient level (which is also at the slide level for many benchmarks), except for CCRCC. Leave-one-out cross validation may lead to overfitting, and this is become more concerning here because the authors use a complex approach which can easily be overfitted on to the small number of datapoints at the slide level.
> > ```
> >
> > Respond: We here present the comparison results using STImage-1K4M[1]. We create benchmarks by selecting cancer samples for each organ and randomly splitting the train/validation/test sets (8:1:1) at the slide level. The statistics are shown below:
> >
> > |  | breast | brain | skin | mouth | stomach | prostate | colon |
> > | --- | --- | --- | --- | --- | --- | --- | --- |
> > | #samples | 189 | 152 | 16 | 16 | 12 | 7 | 4 |
> >
> > The model predicts the top 50 highly variable genes, with Pearson correlation used as the evaluation metric. Note that we only include organs where the models achieve significant correlations (at least >0.1). All results are reported based on 5 different random seeds, and all models use UNI as the image encoder.
> >
> > |  | UNI | BLEEP | TRIPLEX | STFlow |
> > | --- | --- | --- | --- | --- |
> > | breast | 0.395(.000) | 0.244(.040) | 0.466(.010) | 0.428(.025) |
> > | brain | 0.343(.001) | 0.299(.008) | 0.338(.005) | 0.367(.021) |
> > | skin | 0.137(.001) | 0.160(.023) | 0.163(.005) | 0.133(.022) |
> > | mouth | 0.173(.000) | 0.187(.030) | 0.121(.005) | 0.162(.018) |
> > | prostate | 0.217(.000) | 0.271(.024) | 0.267(.042) | 0.313(.010) |
> > | stomach | 0.164(.003) | 0.188(.033) | 0.192(.024) | 0.283(.029) |
> > | colon | 0.276(.003) | 0.301(.015) | 0.243(.003) | 0.288(.000) |
> > | Avg. | 0.243 | 0.235 | 0.255 | 0.282 |
> >
> > The results demonstrate the effectiveness of STFlow, where our method can achieve over 10% improvement. The complete benchmark will be included in our updated manuscript.
> >
> > [1] STimage-1K4M: A histopathology image-gene expression dataset for spatial transcriptomics. NeurIPS2024.

---

> > > ### Author Response · Authors · 2024-12-02
> > >
> > > Thank you once again for your valuable comments on our paper! As the deadline is approaching, we may not be able to respond after that. Please let us know if you have any further questions or concerns. We are committed to addressing your feedback.

---

> > > > ### Author Response · Authors · 2024-12-03
> > > >
> > > > We sincerely apologize for the repeated follow-up. With the rebuttal phase concluding in a few hours, we want to ensure that we have thoroughly addressed all your concerns.
> > > >
> > > > If there are any additional points needing clarification, please let us know. Thank you for your time and thoughtful consideration!

---

> ### Comment · Reviewer_qSWS · 2024-12-03
>
> I think the authors for the response, and have addressed some of my concerns regarding benchmarking. However, I remain concerned with:
> - the author's approach is not quite agnostic to foundation models (FMs), as it requires a new ST prediction module for each FM. Changes in FM leads to a much more significant performance boost compared to changes in the ST prediction module, and that suggests that the FM may be doing the heavy lifting. I encourage the authors to explore methods to fine-tune the patch/spot level encoders as well.
> - Performance boost over baseline TRIPLEX is marginal.
>
> Furthermore, upon reading other reviewer's comments, I concur with their concerns on the lack of novelty, specially with the use of the ZINB distribution, where the ablation studies shows that the ZINB's contribution is limited.
>
> Therefore, I maintain my score of marginally below the acceptance threshold.

---

> > ### Author Response · Authors · 2024-12-03
> >
> > Thanks for your response and we would like to answer your questions point by point.
> >
> > - It is worth noting that our approach consistently brings significant improvements across different foundation models (FMs). For example, STFlow(Ciga) achieves 0.361 compared to Ciga’s 0.305, and STFlow(UNI) achieves 0.419 compared to Ciga’s 0.347. These notable improvements are orthogonal to whether or not the patch encoder is fine-tuned. However, we agree that fine-tuning the patch encoder is a meaningful direction and plan to explore it further.
> > - On the new benchmark we provided, the improvement of STFlow over TRIPLEX is non-trivial (0.282 vs. 0.255). Additionally, we offered TRIPLEX's performance with UNI, which further demonstrates that STFlow consistently outperforms TRIPLEX.
> >
> > Regarding the ablation study on ZINB distribution, we have already provided the explanation which has addressed the other reviewers’ concerns: comparing the average Pearson correlation across datasets might somewhat **underestimate the contribution of flow matching** since there are significant scale differences between datasets (e.g., the model achieves 0.707 on SKCM but only 0.128 on HCC). Here, we present the relative improvement of the model with flow matching on each dataset, where flow matching can provide around 5% improvement on UNI, which is non-trivial.
> >
> > | relative improvement% | Average | IDC | PRAD | PAAD | SKCM | COAD | READ | CCRCC | HCC | LUNG | LYMPH_IDC |
> > | --- | --- | --- | --- | --- | --- | --- | --- | --- | --- | --- | --- |
> > | UNI | 5.04% | 0.68% | 5.79% | 2.42% | 3.06% | 2.01% | 3.84% | 3.39% | 24.27% | 0% | 4.81% |
> > | Gigapath | 1.37% | 0.17% | 0.97% | 1.38% | 0% | 0.61% | -0.38% | -0.60% | 8.69% | 1.51% | 1.32% |
> >
> > We will incorporate all these analyses and discussions in our manuscript. Please let us know if you have another questions.

---

### Meta-Review · Area_Chair_qZv2 · 2024-12-16

**Metareview:**

This work proposes a framework to predict spatial transcriptomics using whole slide images via flow-matching, introducing a new perspective for this field. The research question is timely and important. Experimental analysis shows promising results in spatial transcriptomics prediction.

This paper received 1x marginally above the acceptance threshold and 2x marginally below the acceptance threshold from reviewers. The major concerns raised by reviewers regarding this work centered around the limited novelty, limited experiments, and requirements of detailed explanation of certain aspects (e.g., motivation of ZINB distribution). Although the authors have addressed most of the concerns during the rebuttal, the majority of reviewers still find that novelty is limited and consider this work not suitable for publishment in ICLR. Therefore, rejection is recommended.

**Additional Comments On Reviewer Discussion:**

Reviewers raised questions requiring further clarification of methodology design, experimental settings, and additional analysis. While I appreciate the efforts the authors made during the rebuttal phase, for example, the additional experiments on STImage-1K4M, major concerns regarding methodology novelty remain unaddressed.

---

### Decision · Program_Chairs · 2025-01-22

Reject